# Modeling Myotonic Dystrophy Type 2 Using *Drosophila melanogaster*

**DOI:** 10.3390/ijms241814182

**Published:** 2023-09-16

**Authors:** Marta Marzullo, Sonia Coni, Assia De Simone, Gianluca Canettieri, Laura Ciapponi

**Affiliations:** 1Department of Biology and Biotechnologies “C. Darwin”, Sapienza University of Rome, 00185 Rome, Italy; marta.marzullo@uniroma1.it (M.M.);; 2Department of Molecular Medicine, Sapienza University of Rome, 00161 Rome, Italy; 3Istituto Pasteur Italia, Fondazione Cenci Bolognetti, 00161 Rome, Italy

**Keywords:** *Drosophila melanogaster*, myotonic dystrophy type 2, DM2, CNBP

## Abstract

Myotonic dystrophy 2 (DM2) is a genetic multi-systemic disease primarily affecting skeletal muscle. It is caused by CCTGn expansion in intron 1 of the *CNBP* gene, which encodes a zinc finger protein. DM2 disease has been successfully modeled in *Drosophila melanogaster,* allowing the identification and validation of new pathogenic mechanisms and potential therapeutic strategies. Here, we describe the principal tools used in *Drosophila* to study and dissect molecular pathways related to muscular dystrophies and summarize the main findings in DM2 pathogenesis based on DM2 *Drosophila* models. We also illustrate how *Drosophila* may be successfully used to generate a tractable animal model to identify novel genes able to affect and/or modify the pathogenic pathway and to discover new potential drugs.

## 1. Introduction

Myotonic dystrophy type 2 (DM2, OMIM 602668) is a multi-systemic autosomal dominant disease that displays a wide spectrum of clinical manifestations, including proximal myotonia, degeneration of muscle fibers, cataracts, defective cardiac conduction, insulin resistance, and other endocrine disorders [1,2].

The genetic basis for DM2 is an unstable CCTG repeat on chromosome 3q21, in the first intron of the cellular nucleic acid-binding protein (*CNBP*) gene, also named *ZNF9* (zinc finger protein 9; [3]). The cause for the unstable expansion is unknown; however, it is clear that the expanded DM2 alleles are strongly variable, with significant increases in length over time [4]. The size of the (CCTG)n repeat is below 30 repeats in normal individuals, whereas in DM2 patients, it is between 75 CCTG and 11,000 repeats [3,4]. The typical onset of DM2 is in adulthood and has variable manifestations, such as early onset cataracts (less than 50 years of age), various grip myotonias, thigh muscle stiffness, muscle pain, and weakness in the flexors of the fingers. These complaints often appear between 20 and 50 years of age [2].

The muscular defects observed in DM2 represent the predominant manifestation of the disease and encompass muscle weakness, myotonia (an inability of muscles to relax after contraction), and muscle atrophy over time. The muscles primarily affected by the condition tend to be those closer to the body (proximal muscles) [5]. The severity and the specific muscle groups involved can vary across individuals [5]. Furthermore, being a multi-system disorder, DM2 can affect organs and systems beyond the muscular system. The central nervous system involvement in DM2 has been the subject of extensive investigation in recent years, although several questions remain unanswered. Neurodegeneration evidence exists in DM2, especially in certain brain regions. Individuals with DM2 have reported cognitive impairments, including problems with attention, memory, and executive functions. These cognitive issues underscore central nervous system involvement and imply a neurodegenerative aspect of the disease [6]. While most of these insights have originated from clinical observations, comprehensive studies to examine the nervous system’s contribution to pathology and the interplay between neuronal and muscular degeneration have not been elucidated yet.

To characterize molecular mechanisms underlying DM2 pathogenesis, different vertebrate and invertebrate animal models have been successfully generated. Interestingly, *Drosophila* has emerged as a very reliable model for studies on DM2 since the observed phenotype is highly reminiscent of human disease. 

In this review, we will describe the principal tools used in *Drosophila* to study and dissect molecular pathways related to muscular dystrophies and summarize the main findings on DM2 pathogenesis based on DM2 *Drosophila* models. Finally, we will illustrate how *Drosophila* may be successfully used to generate a tractable animal model to identify novel genes able to affect and/or modify the pathogenic pathway and to discover new potential drugs.

## 2. DM2 Pathogenesis

The pathogenic mechanism of DM2 is still not fully understood. There are three main hypotheses of how the *CCTG* repeat expansion results in the disease’s manifestation (Figure 1). 

### 2.1. CNBP Protein Loss of Function

According to some studies [7,8,9,10,11,12], the CCTG expansion localized in the first intron of *CNBP* affects its expression *in cis* by forming dsDNA secondary structures that alter transcription [12] or by inducing nuclear sequestration of the expanded transcripts [7], leading to haploinsufficiency; indeed, mice carrying homozygous or heterozygous deletion of the *CNBP* allele develop clinical manifestations strongly reminiscent of DM2 myopathy [11]. Similarly, the silencing of *CNBP* from *Drosophila* muscle tissues causes severe locomotor defects that can be fully rescued by reconstitution with either *Drosophila CNBP* or by its human counterpart [13]. However, while some studies reported that CNBP protein levels are significantly reduced in muscle of DM2 patients, other works failed to observe such reduction [7,8,9,10,11,14], most likely as a consequence of the limited sample sizes and the variability of the disease.

CNBP is a highly conserved ssDNA-binding protein [15] involved in the control of transcription by binding to ssDNA and unfolding G-quadruplex DNAs (G4-DNAs) in the nuclei, or translation, by binding to mRNA and unfolding G4-related structures in the cytosol [8,16,17,18,19,20]. Thus, CNBP protein deficiency can also affect CNBP targets correlating with the pathogenesis of DM2. In line with this view, we have recently demonstrated that CNBP is involved in polyamine biosynthesis by regulating the translation of ornithine decarboxylase (ODC; [13]), a key regulator of the metabolism of polyamines. 

### 2.2. Toxic Gain of Function mRNA from Expanded Repeats

The CCTG expansion can be transcribed bidirectionally, resulting in the generation of both a sense and an antisense transcript [21,22]. The accumulation of these transcripts can give rise to a toxic expanded RNA that has been proposed to have three main *gain-of-function* pathological mechanisms: (1) formation of toxic repeated RNA foci; (2) splicing defects related to defective functions of RNA-binding proteins, such as the muscleblind-like proteins (MBNL1-3) and CUG-binding protein 1 (CUG-BP1) [23,24]; (3) a recently discovered retention of the long intron 1 in *CNBP* mRNA. Retention of intron 1 has been found in different DM2 patient-derived cells, suggesting that CCUG expansions can have an inhibitory effect on *CNBP* pre-mRNA splicing by altering the RNA structure and/or the access of splicing factors to intronic regulatory regions [25].

### 2.3. Tetrapeptide-Repeat Rrotein (TPR)-Mediated Toxicity

Intronic *CCUG* expansion in the *CNBP* mRNA can undergo non-canonical Repeat Associated Non-AUG (RAN) translation [21,26,27], producing two different tetrapeptide repeated protein TPRs (LPAC and QAGR) that disrupt cellular homeostasis [21]. The two TPRs are produced by the bidirectional translation of the *CCUG* expansion, producing the LPAC tetrapeptide (leucine–proline–alanine–cysteine) in the sense direction and the QAGR tetrapeptide (glutamine–alanine–glycine–arginine) in the antisense direction. Both LPAC and QAGR have been found to be accumulated in brain biopsies from DM2 patients and seem to be responsible for at least some of the neurological features in people affected by myotonic dystrophy type 2 [21,28]. 

Each of these three potential mechanisms of toxicity is likely to contribute to disease initiation and progression; however, it is unclear to what extent each of them contributes to the development and the clinical manifestations of the disease and how they interact with each other or whether they act synergistically [22]. It has recently been proposed that in the early stage of the disease, the main DM2 pathogenic mechanisms are *CNBP* haploinsufficiency and RNA toxic gain-of-function, while later toxic mRNAs are transported to the cytoplasm, where RAN translation occurs, leading to the production of toxic peptide and to a worsening of the phenotype [22]. 

In addition to DM2, there is another form of myotonic dystrophy, DM type 1 (DM1, Steinert’ disease, MIM 160900), caused by an expansion of CTG repeats in the 3′ untranslated region of the DM protein kinase (*DMPK*) gene [29]. DM1 and DM2 display several similarities in clinical features, although DM2 lacks a congenital or early onset form.

The finding that these two distinct mutations cause largely similar clinical syndromes has highlighted that they share similar molecular mechanisms [30]. However, additional pathogenic mechanisms like changes in gene expression, microRNA, epigenetic modifications, protein translation, and metabolism may contribute to disease pathology and clarify the phenotypic differences between these two types of myotonic dystrophies [1].

## 3. *Drosophila melanogaster* as a Tool to Study Neuromuscular Disorders

*Drosophila melanogaster* is a powerful animal model that can be used for genetic studies of human diseases. Fruit flies share around 75% of human disease-related genes [31,32], a similarity that makes *Drosophila* an excellent in vivo model system capable of revealing novel mechanistic insights into human disorders, providing the foundation for translational research and the development of therapeutic strategies [33,34]. In recent years, *Drosophila* has emerged as an excellent model organism for human neurodegenerative and neuromuscular disorders [35]. The fruit fly offers multiple advantages for the investigation of the molecular mechanisms of this kind of disease. In particular, the large progeny and the short life cycle allow for rapid study of the effects of genetic mutations on the neuromuscular system over the course of life [33,34,35]. Several biological assays have been developed for analyzing the possible role of genetic and/or chemical modifiers in the pathogenesis of the diseases (Figure 2). 

In order to determine movement defects in *fly* models for neuromuscular diseases such as DM2, it is possible to evaluate individual locomotor capabilities in wandering larvae or adult flies. One of the most common and convenient bioassays is the analysis of larval peristalsis. A peristaltic wave is a muscle contraction that propagates along the animal body and involves the simultaneous contraction of the left and right side of each segment, allowing larval movement [38]. To analyze the motility of *Drosophila* larvae, it is possible to quantify different parameters such as the number of larval peristaltic waves performed in 1 min, the distance covered by each larva in the time unit (speed of larval locomotion), and the duration of the peristaltic wave (Figure 2A; [13,39]. To assess movement capabilities in *Drosophila* adults, a widely used locomotion assay is the climbing assay, in which locomotion performances can be assessed using the fly negative geotactic response. In this test, an equal ratio of males or females of the desired ages are placed into a conical tube, flies are tapped down to the bottom of the tube and their subsequent climbing activity is quantified as the percentage of flies reaching the top of the tube in 10 s (Figure 2B) [40,41]. Other parameters that can also be evaluated to measure locomotor capabilities in adult flies are the distance covered by each fly in the time unit (fly speed) or the decrease in locomotor performance on repetition of the test (fatigue) [42,43]. The outcomes derived from these tests are also regarded as indicators of muscle functionality.

The decline in locomotor function is also a prominent feature of aging, and it is evident that aging progressively modifies the physiological balance of the organism, increasing the susceptibility to neuromuscular degenerative diseases [44,45]. However, how aging interconnects with disease-causing genes is not well known. Mutation in disease-related genes in *Drosophila* can also affect the lifespan and accelerate aging; therefore, it is crucial to analyze survival modifiers using the comparison of survival curves. A survival curve is a graphical representation of the proportion of a population that survives over time [46]. The fruit fly is a highly advantageous model organism for studying the mechanisms of aging due to its relatively short lifespan, cost-effective breeding, and large number of progeny [33,34]. To measure longevity in *Drosophila* and to generate a survival curve, groups of flies are maintained under tightly controlled environmental conditions, such as temperature, humidity, and light cycle, and their survival is monitored over time. The survival of the flies is assessed by counting the number of living and dead flies at regular intervals. The data collected from these observations are plotted on a graph where the shape of the curve provides insights into the aging process and whether mutation in specific disease genes affects the lifespan (Figure 2C) [13,35,47,48]. 

Neurodegeneration can also be easily analyzed in *Drosophila* eyes. The compound eye of the fruit fly is composed of about 800 repeating subunits called ommatidia, each of which consists of an ordered hexagonal array of 8 photoreceptor neurons, so precise that it is often referred to as a “neurocrystalline lattice” [49,50,51]. This rigid organization allows us to exactly evaluate the effect of altered gene expression and mutated proteins on the external morphology of the eye and to detect slight alterations in ommatidia geometry due to cellular degeneration ([35] and references therein). Notably, the eye ommatidia array is disrupted when toxic proteins are expressed during development, allowing, for example, the use of an eye roughness assessment to identify modifiers of RAN-translated peptide toxicity [52,53]. Although eye degeneration is not a prominent feature of DM2, the external eye also offers a rapid readout for genetic screens of genes possibly involved in neuromuscular disorders, as the degenerative eye can show disruption of the ommatidial structure, reduced size, and loss of pigmentation, which can easily be viewed using a dissecting microscope. Of note, the morphology of the eye can be dramatically disrupted without compromising the overall health of the fly (Figure 2D). Using eye-specific GAL4 drivers (*GMR*-GAL4), both disease genes or candidate modifiers can be expressed specifically in the eye, and the effects of highly toxic genes or proteins can be assessed in adult flies without lethality concerns.

Once a new gene is identified as an eye neurodegeneration modifier, it is crucial to subsequently evaluate its function in other tissues that might be more characteristically affected in neuromuscular disorders, such as the brain or muscle. The GAL4/upstream activating sequence (UAS) system is a highly potent tool for precise gene expression. It relies on the properties of the yeast GAL4 transcription factor to activate the transcription of targeted genes by binding to UAS cis-regulatory sites. *Drosophila* strains have been genetically modified to incorporate both components, providing a wide array of combinations. This system is versatile and can be utilized for both gene silencing or expression in specific tissues or developmental stages [54].

Neuromuscular diseases, including myotonic dystrophies, are often characterized by muscular defects, including muscle atrophy and myotonia; thus, it is essential to analyze muscle structure and physiology [55]. In this regard, the muscle fillet of *Drosophila* larvae is a commonly used tissue for studying muscle development and function. The larval muscle fillet can be stained using a variety of techniques to visualize muscle structure and specific markers. For example, fluorescent dyes, such as rhodamine phalloidin, can be used to label muscular actin filaments, while antibodies against specific muscle proteins can be used to identify cell types or structural components [36,56]. Confocal microscopy is then used to capture high-resolution images of the muscle fillet, allowing the analysis of muscle structure and function at the cellular and subcellular levels (Figure 2E) [36]. At the functional level, neurophysiological techniques involving the neuromuscular junction (NMJ) can provide insights into the communication between motor neurons and muscles. The NMJ is a specialized synapse connecting a motor neuron to a muscle fiber, leading to muscle contraction. Parameters, such as the number and branching pattern of neuronal connections to the muscle, are often analyzed to evaluate neuromuscular defects in *Drosophila* larval fillets [57]. In addition, electromyography (EMG), a technique based on recording the electrical activity of muscles in response to nerve stimulation [57], can be used to investigate the NMJ activity. To the best of our knowledge, such techniques have not been utilized yet in *Drosophila* models of DM2.

Defects in muscle development and function can also be evaluated in adult flies by analyzing adult flight muscles. To this end, dorsoventral sections of resin-embedded adult thoraces can be analyzed to measure the area of Indirect Flight Muscles (IFM) and evaluate morphological defects (Figure 2F) [37,58].

## 4. DM2 Pathogenesis Using *Drosophila* as a Study Model

### 4.1. CNBP Protein Downregulation

Haploinsufficiency of the *CNBP* gene, consequent to the nuclear sequestration and/or altered processing of expanded pre-mRNAs, has been proposed to play an important role in the pathogenesis of DM2. Mice carrying a heterozygous deletion of the *CNBP* allele show a phenotype strongly reminiscent of DM2: myotonia, increased fiber type variability, cataracts, and cardiac abnormalities [11,59]. Studies on muscle tissues or myoblasts from DM2 patients provided controversial results regarding *CNBP* haploinsufficiency, possibly related to differences in the experimental design; some studies found normal *CNBP* RNA and protein levels in muscle tissues [60,61], while recent findings documented reduced levels and altered splicing of *CNBP* RNA, with corresponding low protein levels in muscle tissues but not in cell cultures [7,11]. Another study showed decreased levels of CNBP protein but not RNA in DM2 muscle cell cultures, suggesting that the pathological expansion could affect the processing, the nuclear export, or the translation of the mutated RNA [62]. 

In line with this, ablation of *CNBP* from *Drosophila* muscle tissues has recently been shown to cause severe locomotor defects, which can be fully recovered by reconstitution with *Drosophila* CNBP or its human counterpart [13]. The CNBP-dependent locomotor phenotype in *Drosophila* is linked to the ability of CNBP to control polyamine content by regulating the translation of ornithine decarboxylase (ODC; [13]). ODC is a key regulator of the metabolism of polyamines (putrescine, spermine, and spermidine), small intracellular polycations that control essential cellular functions, such as cell growth, viability, replication, translation, differentiation, and autophagy [63,64,65,66]. Because of their critical role, the intracellular concentration of polyamines is tightly regulated; thus, *CNBP* loss of function has a strong impact on the processes related to these molecules. Of note, muscle biopsies obtained from DM2 patients showed reduced levels of both CNBP and its translational target ODC compared to healthy individuals, as in the DM2 fly model. Consistently, the content of the ODC metabolite putrescine was also significantly reduced in DM2 patients, indicating that polyamine synthesis might indeed be downregulated in the human disease context [13]. 

Remarkably, it was observed that polyamine feeding rescues the locomotor defects in the dystrophic fly model, suggesting a potential novel therapeutic avenue to treat DM2 patients. These findings highlight how *Drosophila* represents an excellent model to study the DM2 pathogenic mechanisms related to *CNBP* loss of function, and to identify possible new therapeutic strategies.

### 4.2. Toxic Gain of Function of RNAs—Bi-Directional Antisense Transcription

The expansion of the CCUG repeat in intron 1 of CNBP results in the synthesis of a long pre-mRNA. This toxic mRNA triggers a gain-of-function mechanism that elicits the formation of nuclear foci; the sequestration of splicing factors, such as MBLN, with consequent splicing defects; and the retention of CNBP intron 1 [25,67].

In order to investigate the pathogenic mechanism of CCUG repeat expansions in an animal model of DM2, flies expressing pure, uninterrupted CCUG repeat expansions, ranging from 16 to 720 repeats in length, have been generated [68]. Transgenic expression of the expanded CCUG repeats with an eye-specific driver *GMR-GAL4* leads to abnormal pigmentation and a rough eye surface, indicative of disruption of the ommatidial structure and neurodegeneration. The severity of the phenotype was dependent on the length of the CCTG repeat. Similarly, the specific expression of CCUG-expanded RNA in muscle using the *How^24B^-GAL4* driver leads to the formation of toxic ribonuclear foci in the cytoplasm of muscle cells. These results indicate that this DM2 fly model recapitulates key features of human DM2, including RNA repeated-induced toxicity, ribonuclear foci formation, and changes in alternative splicing dependent on MBNL [68]. Interestingly, the levels of CNBP protein are not mutated in these flies, suggesting that CNBP haploinsufficiency is not related to the sole quadruplet expansion but rather to the genetic mutation occurring in the proper context of the human gene [13].

Moreover, the expression of (CCUG)_106_ repeats in the *Drosophila* eye has been shown to trigger a strong apoptotic response [69]. Inhibition of apoptosis through chemical compounds rescued the retinal disruption phenotype, underlying the power of this DM2 *Drosophila* model as a tool for drug screening. Indeed, in a recent study, 3140 small-molecule drugs from FDA-approved libraries were screened through lethality and locomotion phenotypes using the DM2 *Drosophila* model expressing (CCTG)_720_ repeats in the muscle. Ten effective drugs that improved both the survival and locomotor activity of DM2 flies have been identified, uncovering potential drug targets that may mitigate the progression of the disease [36].

A common feature of both DM1 and DM2 is the ability of CUG- and CCUG-expanded RNA, respectively, to form secondary structures and sequester RNA-binding proteins forming nuclear foci [70]. CCUG repeats tend to bind MNBL with higher affinity than CUG and to form bigger foci. However, DM2 patients generally experience a milder phenotype than DM1 patients. 

To explore this paradox and address divergent aspects of pathology in DM1 and DM2, novel *Drosophila* models expressing the respective CUG- and CCUG-expanded RNA in skeletal and cardiac muscle (using the muscle-specific driver myosin heavy chain *Mhc-Gal4* or the cardiac-specific driver *GMH5–Gal4*), have been generated and evaluated [58]. The expression of either CUG or CCUG-expanded repeats has been shown to sequester MBLN in ribonuclear foci in both muscle and cardiac tissue and that, as a consequence, MBNL-dependent splicing was altered. Interestingly, the expression of autophagy-related genes (*Atg4*, *Atg7*, *Atg8*, *Atg9*, *Atg14*) has been found to increase in the muscular and cardiac tissues of both DM1 and DM2 model flies [58]. Physiologically, expression of CUG- or CCUG-expanded RNA in the muscles caused muscle degeneration with consequent reduced muscle area, diminished survival, and decreased locomotor performance [58]. The two DM1 and DM2 fly models represent excellent animal models to investigate the clinical differences between these two human diseases, to increase knowledge about their pathogenesis, and to improve the development of new treatments. 

The important role of MBNL1 in both DM1 and DM2 pathogenesis is also supported by the evidence that cardiac overexpression of *Mbnl*, the *Drosophila* MBNL1 ortholog, is sufficient to rescue the heart dysfunctions and the reduced survival observed in the DM1 and DM2 fly models [37]. Interestingly, it has also been found that the CCUG repeated RNA is bound by rbFox1, an RNA-binding protein involved in the regulation of different phases of RNA physiology [71,72,73]. Differently from MBNL, rbFox1 preferentially associates with the CCUG repeats and not with CUG repeats and is sequestered in ribonuclear foci. Overexpression of rbFox1 has been shown to rescue both the muscular atrophy and locomotion ability of flies bearing the CCUG repeat expansion, demonstrating the importance and specific role of this protein in the pathogenesis of DM2.

### 4.3. RAN Translation-Protein Toxicity

The use of *Drosophila melanogaster* as a model organism has also been instrumental in studying the toxicity of repeat-associated non-AUG (RAN) proteins, which are produced by non-canonical translation of abnormal repeat expansions in various genetic disorders, including myotonic dystrophy type 2 [21,26]. Through RAN translation, a protein is synthesized from a repeated nucleotide sequence that does not contain an AUG codon. The repetitive peptides are the result of RAN translation initiating at different sites within the repeat expansion, leading to the generation of different aminoacidic sequences depending on the reading frame [26].

CCTG expansions in DM2 have been shown to be bidirectionally expressed; thus, transcribed CCUG-repeated RNAs can be translated in two different tetrapeptide repeats: LPAC, leucine–proline–alanine–cysteine in the sense direction or QAGR, glutamine–alanine–glycine–arginine in the antisense direction [21]. These tetrapeptide products are repetitive in nature and can have aberrant biochemical behavior, leading to their accumulation inside the cells. Of note, the accumulation of these toxic RAN products has been implicated in the pathogenesis of DM2 and the associated cellular dysfunction in different tissues [21]. Interestingly, LPAC and QAGR peptide-mediated toxicity seem to be independent of RNA gain of function in DM2 pathogenesis [21]. In DM2 brain autopsy samples, LPAC proteins have been found in the gray matter, including neurons, astrocytes, and glia, and QAGR proteins have been found in the white matter [6,21]. 

The *Drosophila melanogaster* model of DM2-CCTG RAN-translation has not been reported yet. However, *Drosophila* has been successfully used in a model of amyotrophic lateral sclerosis and frontotemporal dementia to dissect the pathogenic mechanisms of the disease [74,75,76,77]. This example supports *Drosophila* as an effective system for the study of RAN-dependent protein toxicity in neuromuscular degenerative diseases. Similar approaches could be set up to characterize the toxic contribution of RNAs and RAN tetrapeptides to the onset and progression of DM2 pathogenesis.

## 5. Conclusions

In conclusion, *Drosophila melanogaster* has proven to be a valuable animal model for studying myotonic dystrophy type 2 (Table 1). Although DM2 is a human-specific disorder, researchers have successfully utilized fruit flies to gain insights into the underlying mechanisms of the disease. The ability to dissect the different pathogenic mechanisms in DM2 fly models has provided evidence that both loss of function of *CNBP* and RNA toxic gain of function of the CCUG repeat contribute to pathology. Studies on *Drosophila* CNBP loss of function showed that the CNBP-dependent locomotor phenotype is linked to the ability of CNBP to control polyamine content by regulating the translation of ODC. Remarkably, polyamine feeding rescues the locomotor defects in this fly model, suggesting a potential novel therapeutic avenue for treating DM2 patients.

The CCUG repeat toxicity also plays a crucial role in inducing DM2 disease through the sequestration of MBNL1 and rbFox1 factors and the formation of ribonuclear foci in muscle cells. Interestingly, it has been demonstrated that overexpression of Mbnl or rbFox1 in *Drosophila* is capable of rescuing both muscular atrophy and the locomotion ability of flies bearing the CCUG repeat expansion. Furthermore, a recent study identified ten effective drugs that improved both the survival and locomotor activity of the DM2 *Drosophila* model expressing (CCUG)_720_ repeats in the muscle, uncovering potential drug targets that may mitigate the progression of the disease.

Ultimately, *Drosophila* models have significantly accelerated the discovery of deregulated genes and pathways in DM2, including regulators of autophagy and apoptosis. 

The possibility to knock down the CNBP gene or to express the CCTG repeated RNA in specific fly tissues allowed us to selectively recapitulate the distinct DM2-associated molecular alterations and the corresponding phenotypes. Thus, fly DM2 models have been pivotal to discern the individual contribution of the different pathogenetic mechanisms to the onset and progression of the disease. 

The examples reported above demonstrate how *Drosophila* has been instrumental in identifying potential therapeutic targets for DM2. The ability to manipulate genes, observe phenotypic effects, and conduct large-scale genetic screenings in *Drosophila* has provided, and will surely continue to do so, additional valuable insights in the understanding of this complex disease that still lacks a resolutive treatment. However, while *Drosophila* has improved our comprehension of DM2, it is important to acknowledge that it cannot fully replicate the complexity of the human disease. Thus, further investigations using complementary model systems and clinical studies are essential for a full understanding of DM2 and the development of effective therapies. 

## Figures and Tables

**Figure 1 ijms-24-14182-f001:**
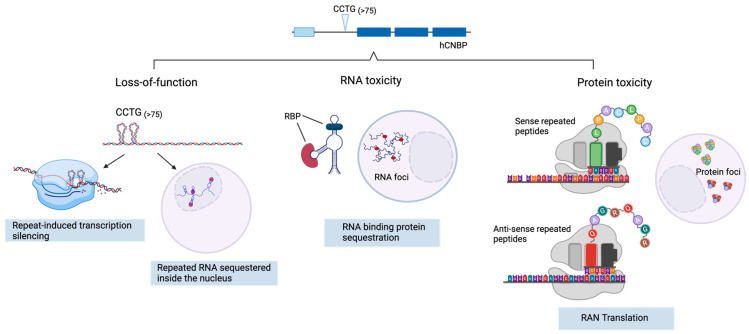
Possible molecular consequences of CCTG nucleotide repeat expansion in the *CNBP* gene. *Loss of function:* expansion of the repeats form dsDNA secondary structures that can elicit transcriptional gene silencing, resulting in partial or complete loss of the native protein encoded by the *CNBP* gene. Transcribed repeated RNAs can also fold into complex structures that are sequestered into the nucleus, resulting in haploinsufficiency. *RNA toxicity*: transcribed CCUG repeated RNAs aberrantly interact with and sequester RNA-binding proteins, forming toxic RNA foci. *Protein toxicity*: non-coding RNA repeats, lacking the canonical AUG translation initiation codon, undergo non-canonical repeat-associated non-AUG (RAN) translation, thus producing LPAC (sense) and QAGR (antisense) toxic tetrapeptides. Created with BioRender.com.

**Figure 2 ijms-24-14182-f002:**
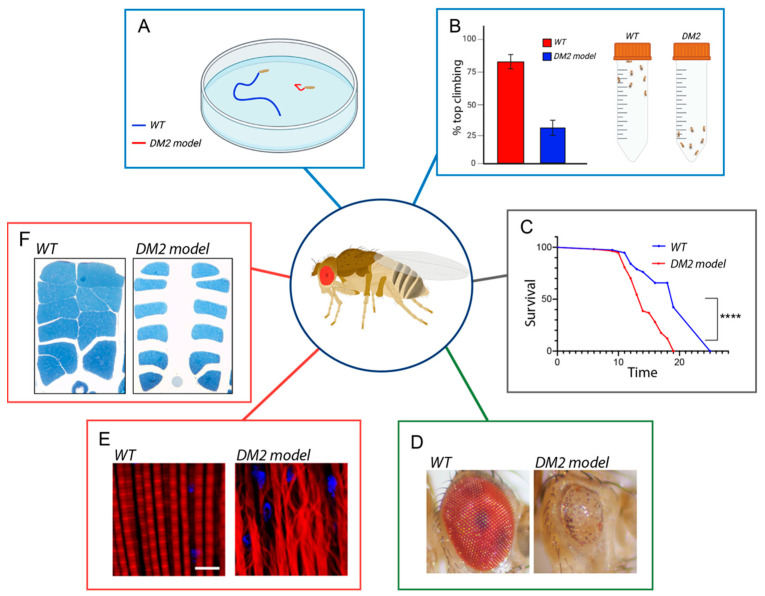
Examples of powerful assays used to assess neuromuscular degeneration and dysfunction in DM2 *Drosophila* models. Several behavioral tasks, such as (**A**) larval crawling and (**B**) adult climbing, allow monitoring the locomotor activity during *Drosophila*’s life. (**C**) *Drosophila* lifespan assays are useful to follow the time course of neuromuscular degeneration and might be used as a readout for genetic screens, example of statistical significance **** *p* < 0.001 determined by long-rank test. (**D**) Disease genes can be expressed in the eye using specific *GAL4* drivers to analyze neurodegeneration. The external eye offers a rapid readout, as the degenerative eye can show disruption of the stereotyped organization of ommatidia, leading to a rough eye phenotype. This easily observable phenotype enables genetic screens aimed at identifying modifiers (enhancers or suppressors) of eye alteration. (**E**) Analysis of larval muscles (adapted from [36]) or (**F**) adult flight muscles (adapted from [37]) are other important tools used for assessing defects in the development and function of the muscles associated with muscular dystrophies. When not specified, they are our original images.

**Table 1 ijms-24-14182-t001:** Reference table of *Drosophila melanogaster* DM2 models.

Pathogenic Mechanism	*Drosophila* Model	Affected Tissue	Phenotype	Ref
Loss of function	*UAS-CNBP^RNAi^*	Muscle (*c179-G4; How^24B^-G4; Mef-G4, Mhc-G4*)	LocomotionClimbing	[13]
RNA toxicity	*UAS-CCTG_16_*	Eye (*GMR-G4*), Muscle (*How^24B^-G4*), Nervous system (*elav-G4*)	Eye degeneration	
*UAS-CCTG_200_*	Muscle RNA foci	[68]
*UAS-CCTG_475_*		
*UAS-CCTG_525_*		
*UAS-CCTG_700_*		
*UAS-CCTG_720_*		
		Missplicing	[69]
*UAS-CCTG_106_*	Muscle (*Mhc-G4*)Eye (*GMR-G4*)	Apoptotic response	
*UAS-CCTG_480_*		
		Autophagy	
*UAS-CCTG_20_*	Muscle (*Mhc-G4*)Heart (*GMH5-G4*)	Muscle degeneration	[58]
*UAS-CCTG_1100_*	Reduced survival	
			Locomotor defects

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
