# Peer review of "Modeling Myotonic Dystrophy Type 2 Using Drosophila melanogaster"

_ijms, 2023, doi:10.3390/ijms241814182_

Round 1

Reviewer 1 Report

In their manuscript titled "Modeling Myotonic Dystrophy type 2 using Drosophila melanogaster", Marzullo et al. provide a comprehensive review of the experimental approaches and advantages that Drosophila offers to study DM2.

The article is interesting, informative and well written, and I therefore believe that it is suitable for publication in the International Journal of Molecular Sciences with only minor revisions.

Major points

1) Fig. 1. The cartoon depicting the loss-of-function of CNBP induced by transcriptional silencing shows stem loop structures that are reminiscent of RNA hairpins... but seem to be made of dsDNA... Is this what was meant by the authors. Does dsDNA form hairpin structures like the ones shown in the cartoon...? Or were those hairpins supposed to represent RNA stem-loops ? If that is the case, the cartoon should not show a double-helix (and the repeat sequence should be CCUG, not CCTG). I find the figure confusing.

2) For RAN translation, there is a requirement for antisense transcription. Even though there's a fleeting mention to antisense transcripts later in the manuscript, as a reader I found myself wanting to learn more about this mechanism. While the LPAC and QAGR peptides expected from RAN translation can be found in samples, what do we know about the accumulation of a.s. transcripts? Does a.s. transcription of this locus happen during homeostasis as well, or is it another phenotype associated with the amplification of the CCTG repeat? And if so, do we know if there's some type of threshold over which a.s. transcription of CNBP "kicks in"?

3) In Fig. 2, panels E and F are indicated as adapted from references. What about panels A-D? Are these adaptations of real data? And if so, what references were they adapted from? This should be indicated in the figure legend, like it is for E and F.

4) L199-202: I find this passage misleading, as it implies that eye abnormalities are a readout of neuromuscular disorders. The wording seems off. I know the authors understand that eye malformations are just a readout for, in this case, neurodegeneration and, more generally, morphogenetic and cellular phenotypes. But the wording "...for genetic screens on neuromuscular disorder..." Somehow imply that the fly eye abnormalities are a good model for neuromuscular systems (which is not). I'd recommend some more careful re-phrasing of this sentence.

Minor points, typos, small edits

L29: a closing parenthesis is missing after [3].

L32: I'd recommend changing "Myotonic dystrophy type 2 typical onset is in adulthood" for "The typical onset of DM2 is in adulthood".

L30-133: "Drosophila" is not italicized a couple of times. And "in vivo" (L130) should be italicized as well. I did not pay close attention to the rest of the manuscript, to see if there were other misses. But Drosophila and Drosophila melanogaster should be consistently italicized throughout the manuscript (as well as in vivo, in vitro, per se, etc...)

L162: I would recommend changing "To analyze the ability and efficiency of Drosophilaa larvae to move" for "To analyze the motility of Drosophila larvae".

L171: "can be also evaluated" >> "can also be evaluated"

L183: "...longevity in Drosophila and TO generate a..." (missing "to")

L196: "[30] and references in it" >> "[30] and references therein"

L221: Missing comma in "To this purpose, dorsoventral...."

L231: Haploinsufficiency of >the< CNBP gene,...

L233: Mice carrying >a< heterozygous....

L246: >The< CNBP-dependent locomotor phenotype...

L265: There's a mention to the bi-directional antisense transcription of the CNBP locus here, but no further elaboration in this section. The a.s. transcripts are mentioned in section 4.3 - but they are not elaborated in enough detail. As I suggest in item #2 above, I think the article would benefit significantly from including a more detail discussion of the mechanisms mediating this bi-directional transcription, etc.

L280: dependent by MNBL >> dependent >on< MBNL

L300: "in" is repeated - ...RNA in in skeletal...

L304: "as consequence" >> "as >a< consequence" or "consequently"

Author Response

Please see the point by point response in the attachment

Reviewer 2 Report

The manuscript Modeling Myotonic Dystrophy type 2 using Drosophila melanogaster by Marzullo et al reviews the current contribution of fly models to our understanding of Myotonic Dystrophy type 2.  Overall, the manuscript is well written and provides a succinct account of the contributions of Droosphila to the disease.  A few minor concerns are noted:

1. in Figure 1: the left most panel is not easy to understand. I suggest showing the expansion of the repeat rather than an unexpanded one and using arrows to show that the repeat induces repeat induced transcription silencing and Repeat RNA sequestering instead of the use of repressive symbol suggesting that the CCUG blocks them.  Change the CCUG(n) to CCUG(>75) to represent that the expansion is the cause.

2. Line 113-114. It is unclear what is meant by a deterioration of the phenotype.  Please explain more clearly.

3. The authors often discuss neurodegeneration in the context of assays (line 149, 190, 207, 275) but do not really introduce the connection between dying neurons (neurodegeneration) and defects caused by the CCTG expansion in CNBP.  For example, the opening paragraph that describes the disease fails to mention neurological manifestations at all.  Is there a neurodegenerative component or is the disease solely caused by muscle problems?  The lack of neuronal stimulation could cause muscle deterioration, or the muscle deterioration could cause neuronal problems or they could both happen independent of each other.  Which is the case for DM2 is not clear.

 4. In the section: 3. the authors seem to cherry pick a few of the assays as good candidates for a screening tool, when in fact, most of the assays mentioned have been used as successful screening tools.  A broader context should be considered, as forward genetic screens are one of strongest tools in Drosophila.

5. Also in section 3: The daily activity monitor assay is also used in this field to great effect (for example author’s citation #10)

6. Line 212-213: the authors state : “muscle fillet of Drosophila larvae is a commonly used tissue for studying muscle development and function” yet do not describe any functional assays, only structural ones.  Is neurophysiology of the neuromuscular junction a tool to possibility evaluate the function of the NMJ and does it relate to DM2?  What about neurophysiology of the giant fiber to measure the responsiveness of the flight muscles?    Either way, if the authors suggest the larval filet as a method to study the function of the muscles then they should provide some examples for functional studies, even if they have not been done in models of DM2.

7. Also in section 3: I am surprised that the molecular tools of Drosophila receive so little attention in this section.  The only molecular tool mentioned is the GMR-GAL4 driving transgenes.  The molecular techniques of Drosophila are one of their greatest advantages and figure into several examples in later sections and in the discussion.  At the least, the authors should better describe the GAL4/UAS system and cite the original paper showing its use in flies (Brand and Perrimon 1993) rather than assume the readers will know its details.  A direct mention of the ease and power of genetic manipulations possible with Drosophila would benefit this section.

 8. Conclusions section:   This section seems a little light and is overly general.  For example, the authors state “researchers have successfully utilized fruit flies to gain insights into the underlying mechanisms of the disease”. Which insights?  Refreshing the audience’s memory here by highlighting 2 or 3 of the most important insights would greatly benefit the section.   Similarly, the authors state: [flies have made significant] “contribution of the different pathogenetic mechanisms”.  Please give an example or two to show the readers this is true. Highlighting some of the rescue experiments presented in the text would be appropriate here, for example.

9. A final minor concern is that the paper lacks the perspective of recent molecular tools being developed and used in the modeling of other human diseases in Drosophila that might advance the field of DM2 research.  It seems to me the authors are missing an opportunity to direct the field forward by including discussion regarding some of the recent advances in molecular manipulation of Drosophila made possible by CRISPR/Cas9 gene editing and other molecular manipulations in Drosophila.  For example, the authors could talk about the potential of the CRIMIC GAL4’s which are emerging as significant tools for the study of human diseases using Drosophila (for references look at Hugo Bellen’s recent work), or, the various emerging collections of fluorescent and epitope tagged (GFP, V5 for example) transgenes and endogenous CRISPR knock-ins that could be used to study DM2. Although to be fair these reagents don’t seem to be available yet for DM2 related genes –for example the CRIMIC is not yet available for CNBP – but the author could look at the Drosophila disease modeling literature and suggest the utility of making/using these and other molecular reagents.  In another example directly knocking expanded CCUG repeats into the endogenous CNBP locus using CRISPR might be an excellent new model. It just feels like their final statement of the introduction: “At last, we will illustrate how Drosophila may be successfully used to generate a tractable animal model to identify novel genes able to affect and/or modify the pathogenic pathway, and to discover new potential drugs” falls a little short without some mention of what can be done in the future to advance the Drosophila models of DM2.  I consider this a minor concern as the scope of the paper is fulfilled without this, but I feel the authors are missing an opportunity, as I said, to help move the field forward by thinking about some of the tools and techniques in Drosophila being applied to other human disease modeling and might benefit DM2. The current loss-of-function stocks for CNBP, for example, are fairly limited with only p-element insertions and UAS RNAi/CRISPR-based Knock down constructs.  The field could greatly benefit from better null alleles of the gene, which could be easily accomplished with modern CRISPR/Cas9 based methods.

Author Response

(The authors gave the same response as above.)
